# Interpreting Viral Deep Sequencing Data with GLUE

**DOI:** 10.3390/v11040323

**Published:** 2019-04-03

**Authors:** Joshua B. Singer, Emma C. Thomson, Joseph Hughes, Elihu Aranday-Cortes, John McLauchlan, Ana da Silva Filipe, Lily Tong, Carmen F. Manso, Robert J. Gifford, David L. Robertson, Eleanor Barnes, M. Azim Ansari, Jean L. Mbisa, David F. Bibby, Daniel Bradshaw, David Smith

**Affiliations:** 1MRC-University of Glasgow Centre for Virus Research, Glasgow G61 1QH, UK; Emma.Thomson@glasgow.ac.uk (E.C.T.); joseph.hughes@glasgow.ac.uk (J.H.); Elihu.Aranday-Cortes@glasgow.ac.uk (E.A.-C.); John.McLauchlan@glasgow.ac.uk (J.M.); Ana.daSilvaFilipe@glasgow.ac.uk (A.d.S.F.); lily.tong@glasgow.ac.uk (L.T.); Robert.Gifford@glasgow.ac.uk (R.J.G.); David.L.Robertson@glasgow.ac.uk (D.L.R.); 2Virus Reference Department, National Infection Service, Public Health England, Colindale, London NW9 5EQ, UK; Carmen.Manso@phe.gov.uk (C.F.M.); Tamyo.Mbisa@phe.gov.uk (J.L.M.); David.Bibby@phe.gov.uk (D.F.B.); Daniel.Bradshaw@phe.gov.uk (D.B.); 3Peter Medawar Building for Pathogen Research, Nuffield Department of Medicine, University of Oxford, Oxford OX1 3SY, UK; ellie.barnes@ndm.ox.ac.uk (E.B.); ansari@well.ox.ac.uk (M.A.A.); david.smith@stcatz.ox.ac.uk (D.S.)

**Keywords:** deep sequencing, virus genomics, hepatitis C virus, variant calling, sequence interpretation, drug resistance, bioinformatics

## Abstract

Using deep sequencing technologies such as Illumina’s platform, it is possible to obtain reads from the viral RNA population revealing the viral genome diversity within a single host. A range of software tools and pipelines can transform raw deep sequencing reads into Sequence Alignment Mapping (SAM) files. We propose that interpretation tools should process these SAM files, directly translating individual reads to amino acids in order to extract statistics of interest such as the proportion of different amino acid residues at specific sites. This preserves per-read linkage between nucleotide variants at different positions within a codon location. The samReporter is a subsystem of the GLUE software toolkit which follows this direct read translation approach in its processing of SAM files. We test samReporter on a deep sequencing dataset obtained from a cohort of 241 UK HCV patients for whom prior treatment with direct-acting antivirals has failed; deep sequencing and resistance testing have been suggested to be of clinical use in this context. We compared the polymorphism interpretation results of the samReporter against an approach that does not preserve per-read linkage. We found that the samReporter was able to properly interpret the sequence data at resistance-associated locations in nine patients where the alternative approach was equivocal. In three cases, the samReporter confirmed that resistance or an atypical substitution was present at NS5A position 30. In three further cases, it confirmed that the sofosbuvir-resistant NS5B substitution S282T was absent. This suggests the direct read translation approach implemented is of value for interpreting viral deep sequencing data.

## 1. Introduction

For some virus species, their highly error-prone replication mechanism produces a population of related genomic variants of the virus within a single infected host individual [1]. Sequencing systems such as Illumina’s platform produce short, relatively accurate nucleotide sections of viral genome, often generating thousands of reads for a given genomic location from a single sample [2]. Such deep sequencing technologies therefore offer methods for understanding the nature of viral intra-host diversity. Whole genome and deep sequencing of virus genomes has been widely applied in basic virology research but has also found applications in clinical contexts such as the detection of drug resistance [3].

Various bioinformatics stages must be applied in the interpretation of viral deep sequencing data. Reads unrelated to the virus genome are removed and low-quality reads removed or trimmed. Following this, we must then construct an alignment: how the reads are arranged relative to each other within the virus genome, accounting for sequence homology. Reference-based alignment or mapping methods such as Bowtie 2 [4], BWA [5], MOSAIK [6], Stampy [7] or Tanoti [8] use one or more reference sequences to guide the alignment of reads. In contrast, de novo assembly approaches such as SPAdes [9] and VICUNA [10] use associations derived purely from the read data itself to propose large genome fragments, avoiding the biases arising from the choice of reference sequence. A drawback of de novo methods is that they may not accurately capture the full genomic structure or diversity, thus, for well-known viruses with high levels of genomic diversity, combinations of de novo assembly and reference-based alignment methods, such as shiver [11], are often used. One aspect common to almost all recent methods in this area is that they output their results in the form of a sequence alignment mapping (SAM) file. The SAM format [12] integrates nucleotide, read quality and alignment data in a single file. It was standardised at an early point in the adoption of deep sequencing, allowing diverse methods to be compared with each other and integrated into processing pipelines for a broad set of applications.

A range of variant-calling methods have been developed to analyse genomic heterogeneity within deep sequencing data. Error rates in short read technologies such as Illumina are low but it can be challenging to distinguish errors from real single nucleotide variants (SNVs) occurring at frequencies comparable to the error rate. Therefore, variant-calling methods such as LoFreq [13] and V-Phaser [14,15] apply statistical techniques to the aligned read data to identify probable SNVs occurring even at very low frequency in the presence of sequencing errors. The focus on low frequency SNVs is critical in applications such as cancer genomics where somatic deviations from the consensus are both rare and of high consequence [16]. However, virus bioinformatics has distinct priorities from fields focused on eukaryote or bacterial organisms with higher replication fidelity [17]. Since viruses typically have a low replication fidelity, there is a higher level of diversity within an infected host and the viral population can be expected to contain many variants.

For reasons of clarity within the research community, virus genome locations can be defined in terms of a standard virus strain with a well-established “master reference” genome. In hepatitis C virus (HCV) for example, codons within viral proteins are numbered relative to the H77 strain [18]. Polymorphisms at these standardised locations are reported with phenotypic associations established experimentally or in clinical trials. The advent of deep sequencing data prompts questions such as: What are the relative proportions of different amino acid residues at a given genome location? What proportion of reads support the presence of a certain sequence motif? What proportion of reads indicate a deletion? However, it is challenging to answer these questions within deep sequencing data, since read alignments do not in general use the reference coordinate space, and a mapping between the two spaces must be established and applied.

The genomes of the virus population may contain multiple nucleotide base variants at different positions within a single codon location. Both Verbist et al. [19] and Döring et al. [20] pointed out that linkage between nucleotide positions is lost when variants are called as SNVs. This linkage must be retained within datasets in order to accurately predict the amino acid residues arising from protein translation. Suppose for example we observe significant levels of both adenine (A) and thymine (T) at the first position of a particular codon location. At the second position, we observe cytosine (C) and guanine (G). With cytosine at the third position, what amino acids are the genomes in the virus population coding for? Without retaining linkage, these observations are consistent with a mix of Threonine (codon ACC) and Cysteine (TGC), or alternatively with purely Serine (AGC/TCC), or with any combination of these amino acids. By retaining linkage, we can accurately select between these interpretations. Haplotype reconstruction methods aim to determine linkage by associating sub-populations of reads as haplotypes. Schirmer et al. [21] found that this was exceptionally unreliable for viral deep sequencing data. However, full haplotype reconstruction may not be necessary for practical applications where the variants of interest are linked within the span of a single read or read pair.

We present a subsystem of the GLUE software package [22] called samReporter, focused on the analysis of aligned deep sequencing viral genome data. It directly processes the SAM file format produced by most methods, and can also process the more compact Binary Alignment Map (BAM) format. The samReporter can be instantiated within an existing GLUE project containing reference sequences and alignments for a given virus. This allows the software to establish a reading frame for coding region reads within a SAM file, and map between the read alignment coordinate space and standardised genome locations. In turn, this facilitates the scanning of reads directly for different classes of sequence pattern such as codons, amino acid residues, indels and motifs. This approach of scanning reads directly has the advantage of retaining linkage and we can report how often combinations of variations appear together on the same viral RNA, certainly within a codon location but also further, at least as far as paired-end read data allows.

We demonstrate the benefits of applying the GLUE samReporter to hepatitis C virus (HCV) deep sequencing data. HCV is a positive-sense single-stranded RNA virus of the family Flaviviridae. Its genome of about 9000 bases codes for a single polyprotein that produces 10 mature viral proteins. HCV affects over 100 million people worldwide and can cause liver disease and cirrhosis. The infection can be treated with a range of direct-acting antiviral (DAA) drugs which inhibit three of the mature proteins: NS3, NS5A and NS5B. Such therapies produce a sustained virological response (SVR) in the vast majority of patients, clearing the virus in around 95% of cases [23]. Notwithstanding this therapeutic success, HCV is proving difficult to treat in certain categories of patients, including “retreatment” patients: those for whom prior DAA treatment has failed. It has been shown both in vitro and in vivo that certain resistance-associated substitutions (RASs) in the viral genome confer resistance to DAA drugs [24].

Vermehren et al. suggested that retreatment patients have RASs in multiple drug target genes and that therefore “genomic resistance testing may be useful to select the optimal combination and treatment duration” for subsequent rounds of drug therapy [23]. Recent guidance suggests that, if deep sequencing is used, observing a RAS in 15% of the virus population may be clinically relevant [25]. Tools aimed at HCV resistance testing such as geno2pheno[ngs-freq] [20] suggest frequencies of 2%, 10% and 15%. Thus, while HCV RAS testing can benefit from deep sequencing methods, moderately-low rather than ultra-low frequencies are of most interest. RAS testing for retreatment patients therefore provides a good case study for a deep sequencing data interpretation system.

## 2. Results

We analysed viral genome diversity within a group of 241 HCV retreatment patients sampled within the United Kingdom. A range of genotypes were represented: Gt1 *n* = 115, Gt2 *n* = 5, Gt3 *n* = 104, Gt4 *n* = 14 and Gt6 *n* = 3. The five most frequent subtypes: were 1a *n* = 98, 1b *n* = 13, 3a *n* = 95, 3b *n* = 6 and 4r *n* = 6. Fourteen other subtypes were represented each by three or fewer patients. In three cases, a subtype could not be assigned.

The samReporter scans aligned reads directly, retaining linkage within reads with the intention of more accurate detection of specific variants. To test the benefits of this approach, we also contrived an alternative method that attempts to capture within-host diversity without retaining linkage.

Besides the four concrete bases A, C, G and T/U, IUPAC notation for nucleic acids, used in the FASTA sequence file format, contains 11 ambiguity codes, covering all possible combinations of more than one base [26]. For example, code S can represent a combination of C and G. Any method which calls SNVs can encode these variants within a FASTA file. This aspect of the encoding is used to capture minor nucleotide variants. Web-based HCV drug resistance interpretation systems such as HCV-GLUE [27] and geno2pheno[hcv] [28] do attempt to interpret ambiguity codes if they appear in the input data. We produced FASTA files with ambiguity codes for each sample using the samReporter nucleotide-consensus command, which produces one IUPAC code for each nucleotide position in the SAM reference coordinate space. Read bases with a Phred quality score of less than 25 were excluded. A “concrete” base (A, C, G or T) was encoded at a given position if it appeared both in at least five individual reads and in 5% of the quality-filtered reads at that location. IUPAC ambiguity codes are then used if multiple concrete bases are to be encoded. We found that FASTA files for all but two samples contained at least one ambiguity code. On average, files contained ambiguity codes which represent two bases at 1.02% of nucleotide positions (std. dev. 1.3%) and codes which represent three bases at 0.0161% of nucleotide positions (std. dev. 0.0475%).

A triplet of concrete bases, i.e., a codon, specifies a single amino acid. If ambiguity codes occur within nucleotide data for a given codon location, multiple distinct codons are present in the underlying data at that location; the precise composition is unknown. For a given ambiguous triplet (possibly containing ambiguity codes), there is set of “possible” amino acids comprising any residues coded by one or more of the possible codons. For example, for the ambiguous triplet YTM, the set of possible amino acids is Leucine (L) and Phenylalanine (F) because the set of codons and their corresponding amino acids are CTA (L), CTC (L), TTA (L) and TTC (F). Additionally, there is a (possibly empty) subset of “definite” amino acid residues, i.e., those that must be coded by at least some of the underlying codons, whatever the composition, under reasonable assumptions. For the ambiguous triplet YTM, every combination of codons which produces the ambiguity codes contains at least some codons for Leucine (L); this is the single definite amino acid. In general, if there is a single ambiguity code encoding two bases within the triplet, there will be one or two definite amino acids, and these will also be the only possible amino acids.

The FASTA files were analysed for “ambiguous” codon locations where the definite and possible amino acid sets were different. This typically occurs when there are two ambiguity codes within a triplet. Such locations present a challenge for drug resistance interpretation systems based on FASTA file inputs. Whereas amino acid residues in the definite set can be inferred to be present in the virus population, the status of amino acids in the possible set but not the definite set cannot be established clearly from FASTA data. We excluded from the analysis degenerate locations i.e., those where the possible set, excluding stops, contained more than five amino acids or where the read depth for the whole codon location was less than 10.

In total, 435 ambiguous locations were found in patients, within all ten viral proteins: Core *n* = 62, E1 *n* = 24, E2 *n* = 205, p7 *n* = 5, NS2 *n* = 14, NS3 *n* = 30, NS4A *n* = 1, NS4B *n* = 9, NS5A *n* = 38 and NS5B *n* = 47. The full set of ambiguous locations is given in the Appendix A. Scaling by the length of each region, this implies that the E2 and Core proteins had a higher rate of ambiguous locations. The drug target proteins NS3, NS5A and NS5B have rates in the lower part of the range.

The current version of HCV-GLUE [27] documents 44 locations associated in the literature with resistance to six DAA drugs in current use: 18 in NS3, 15 in NS5A and 11 in NS5B. These are listed in the Appendix A. Within the FASTA data, we found 10 ambiguous resistance-associated locations in nine patients, six in NS5A and four in NS5B. We resolved these locations using the GLUE samReporter, calculating the frequencies of codons and amino acids by directly analysing reads. Codons were excluded if any Phred base quality was below 25. Amino acid residues were deemed to be present if 5% or more of filtered read codons at the location coded for the residue. The 10 locations are shown in Table 1. HCV-GLUE classifies an amino acid as typical at a location for a given subtype if 10% or more of the GenBank sequences of that subtype contain the residue, these are also shown in the table. In all cases except one (R25, NS5B position 159) the definite residues set (not shown) was empty.

In the cases shown in Table 1, samReporter was able to eliminate many possible residues. Whereas the possible set contained up to four residues, samReporter confirmed that only one or two were actually present at the 5% level. Two resistance locations occur three times each and merit a discussion. For NS5A position 30, in subtype 1a the typical residue is Glutamine (Q). In sample R67, samReporter found Tyrosine (Y) at around 74%, which has not been documented as a RAS but is atypical for the subtype. In samples HCV300 and R164 (subtype 3a) the typical residue is Alanine (A); samReporter found Lysine (K), a well-documented RAS, at levels of 92% and 82% respectively. Thus, in these three cases samReporter confirmed the presence of a RAS or atypical substitution. Substitutions at NS5B position 282 have been strongly associated with resistance to sofosbuvir, particularly the substitution of the typical Serine (S) with Threonine (T). In contrast with NS5A position 30, samReporter was able to eliminate this resistant residue and the other possible atypical residue Cysteine (C); in these three ambiguous cases, only Serine is present, but is actually coded by significantly distinct codons in each case. For sample HCV294, the codons were TCT at 54%, AGT at 25% and AGC at 20%, for sample R25, AGC at 85% and TCC at 15%, and, for sample R36, TCC at 92% and AGC at 7%. The effect of the presence of these diverse codons is to create ambiguity at the nucleotide level. One possible explanation for the codon diversity is that Threonine codons became frequent in the viral population during sofosbuvir treatment, and that following the end of the treatment course the descendants of these virions reverted to Serine, but now coded using diverse alternative codons.

The HCV296 sample is typical in the sense that the size of the BAM file (24.6 MB) was closest to the mean for this dataset, it contains ≈282,000 paired-end reads with an average depth across the HCV polyprotein of ≈3700. To evaluate performance, we ran some samReporter commands on this file using a 2014 MacBook Pro with a 2.5 GHz Intel Core i7 processor and 16 GB of RAM. The samReporter was configured to use up to four CPU cores. The amino-acid command was run to translate reads for the whole polyprotein, producing amino acid residue frequencies at each codon location, without any read filters. Using the auto-align feature with a known target reference sequence, this command took 7.5 s. Using the max-likelihood-placer feature, the command took 29.2 s, with most of the extra time spent in the RAxML-EPA step. See Section 3.2 for details of the samReporter design.

## 3. Materials and Methods

### 3.1. Sequencing Data

Deep whole genome HCV sequencing data was derived from blood samples collected from 241 patients resident in the United Kingdom, who had not achieved virological clearance after previous courses of antiviral therapy. Sequencing was performed using target enrichment on Illumina sequencers at three different institutes: the MRC-University of Glasgow Centre for Virus Research (*n* = 56), the University of Oxford Nuffield Department of Medicine (*n* = 25) and the Virus Reference Department at Public Health England (PHE) (*n* = 160).

The Glasgow library preparation protocol was as follows. RNA was isolated from 200 μL of plasma using the RNAdvance Blood extraction kit (Beckman Coulter, Brea, CA, United States) and collected in 27 μL of water. Following conversion of RNA to double-stranded DNA, libraries were prepared for Illumina sequencing using the KAPA DNA LTP Library Preparation Kit (Roche, Basel, Switzerland), and NEBNext Multiplex Oligos for Illumina (New England Biolabs, Ipswich, MA, United States). Libraries were quantified using Qubit dsDNA HS Assay Kit (Invitrogen, Carlsbad, CA, United States) and size distribution assessed using Agilent TapeStation with D1K High Sensitivity Kit (Agilent, Santa Clara, CA, United States); libraries were normalised according to viral load and mass. A 500 ng aliquot of the pooled library was enriched using SeqCap EZ Developer Probes (Roche), following the manufacturer’s protocol. Following a 14 cycle post-enrichment PCR, the cleaned pool was sequenced with 151-base paired-end reads on a NextSeq cartridge (Illumina, San Diego, CA, United States).

The Oxford libraries were prepared for Illumina sequencing using the NEBNext Ultra Directional RNA Library Prep Kit (New England Biolabs) with 8 μL of RNA extracted from plasma using NUCLISENS easyMAG (bioMérieux, Marcy-l’Étoile, France) and previously published modifications of the manufacturer’s guidelines (v2.0) [29]: omission of heat fragmentation, omission of Actinomycin D at first-strand reverse transcription, library amplification for 15 PCR cycles using custom indexed primers [30], and post-PCR clean-up with 0.85x volume Ampure XP (Beckman Coulter). Libraries were quantified using Quant-iT PicoGreen dsDNA Assay Kit (Invitrogen) and size distribution analysed using Agilent TapeStation D1K High Sensitivity kit. A 500 ng aliquot of the pooled libraries (96 plex) was enriched using the xGen Lockdown protocol (Rapid Protocol for DNA Probe Hybridization and Target Capture Using an Illumina TruSeq or Ion Torrent Library v1.0, Integrated DNA Technologies, Coralville, IA, United States) with equimolar-pooled 120 nt DNA oligonucleotide probes (Integrated DNA Technologies) followed by a 12-cycle on-bead post-enrichment PCR. The cleaned post-enrichment ve-Seq library was quantified by qPCR with the KAPA SYBR FAST qPCR Kit (Roche) and sequenced with 150b paired-end reads on a single run of the Illumina MiSeq.

The PHE library preparation protocol is the laboratory component of a pipeline aimed at clinical use; a manuscript describing the full pipeline is in preparation. RNA was extracted from 350 μL of plasma using the NUCLISENS easyMAG system (bioMérieux). Total eluates were subjected to Turbo DNAse treatment (Thermo Fisher, Waltham, MA, United States) followed by library preparation using KAPA RNA HyperPrep kit (Roche). Libraries were pooled based upon DNA concentration and HCV quantity, assessed using the Quant-iT kit on the Glomax platform (Promega, Madison, WI, United States) and the Qiagen QuantiTect kit with primers and probes from Davalieva et al. [31]. Pools were enriched by hybridisation to a biotinylated probe set (Integrated DNA Technologies, described by Bonsall et al. [32]) followed by further PCR cycles depending upon HCV quantity. The two pools were pooled, again by concentration and HCV quantity. The final pool was quantified using the KAPA SYBR FASTA qPCR kit (Roche) on a PRISM 7500 (Applied BioSystems, Foster City, CA, United States) before being sequenced on a MiSeq using Reagent kit v2 (Illumina).

The Illumina read data were processed into SAM files using different bioinformatics pipelines at the different institutions. At Glasgow, reads were trimmed and filtered using TrimGalore [33] with quality threshold 30 and minimum read length 75. The most appropriate HCV reference sequence was identified via a *k*-mer-based approach, using *k*-mers unique to each genotype [34]. SAM files were generated by mapping against the best-matching HCV reference using Tanoti [8]. At Oxford, de-multiplexed sequence read-pairs were trimmed of low-quality bases using QUASR v7.01 [35] and adapter sequences with CutAdapt version 1.7.1 [36] and subsequently discarded if either read had fewer than 50 bases in its remaining sequence or if both reads matched the human reference sequence using Bowtie version 2.2.4 [4]. Remaining reads were mapped using BWA mem [5] and Stampy [7] against a database of reference sequences, both to choose an appropriate reference and to select those reads which formed a majority population for de novo assembly using VICUNA [10] and finishing with V-FAT [37]. The reads were then mapped back to this assembly using MOSAIK [6]. At PHE, human reads were filtered out from trimmed FASTQ files using SMALT [38], remaining reads were then assembled using VICUNA de novo assembly [10]. Contigs were matched to HCV reference genomes using BLAST [39] and gaps filled using LASTZ [40] to generate a draft assembly. Reads were then mapped to the draft assembly with BWA [5].

The deep sequencing data used in this study has been deposited in the NCBI Sequence Read Archive (http://www.ncbi.nlm.nih.gov/sra), under BioProject accession number PRJNA527067 and experiment accession numbers SRX5528430 to SRX5528670.

### 3.2. GLUE samReporter Design

The GLUE samReporter aims to provide a convenient tool for interpreting viral deep sequencing data. As part of the wider GLUE system [22], it can be used interactively in the command line interpreter or within bioinformatics scripts. Instantiated within a GLUE project for a specific category of viruses such as HCV-GLUE, it can take advantage of certain data objects within that project.

When interpreting viral deep sequencing data, one obstacle is mapping the SAM file coordinate space to a standard codon numbering system. Within the HCV-GLUE project the H77 strain (RefSeq accession NC_004102) is defined as the “master” reference sequence object. The precursor polyprotein and the 10 mature proteins are defined as coding features and their locations are specified on the H77 sequence. A wider set of reference sequence objects is also defined within HCV-GLUE; there are currently over 200 of these, based on the ICTV HCV resource [41]. HCV-GLUE then specifies an unconstrained “master” alignment object containing all these reference sequences, which is used to map their locations to those on the H77 sequence. HCV-GLUE also contains a reference phylogeny of the same set of sequences, computed using RAxML [42].

SAM files for HCV typically map each read to a single coordinate space. To interpret individual reads the samReporter must infer sequence homology (i.e., pairwise alignment) between this SAM file coordinate space and one of the reference sequences defined within GLUE—the “target” reference. The simplest method, to specify that the SAM file coordinate space is identical to that of a specific target reference sequence, is appropriate if one of the project’s reference sequences was used for the SAM file coordinate space. A more flexible “auto-align" method allows GLUE to generate a codon-aware pairwise alignment between the consensus of the SAM file and a selected target reference, using techniques based on BLAST+ [39]. This is appropriate if the SAM virus strain is closely related to the target reference, but importantly, it allows the method producing the SAM file, which may have a de novo element, to construct a coordinate space appropriate for the viral reads. The final and most general and robust method is “max-likelihood-placer”. This allows GLUE to select the target reference itself, by feeding the consensus of the SAM file into the first two stages of the GLUE genotyping pipeline. This consists of incorporating it into the master alignment using MAFFT [43], placing it in the reference phylogeny using RAxML-EPA [42] and selecting as the target the reference sequence with the lowest patristic distance from the SAM consensus. The auto-align method is then used to generate the homology. The master alignment will also typically act as the “linking” alignment, providing a mapping between the target reference and the master reference. The result of this process is then a chain of pairwise homology relationships, as shown in Figure 1, from each individual read to the SAM file coordinate space, to the target reference sequence and finally, via the linking alignment, to the master reference sequence.

The samReporter offers a range of GLUE commands for interpreting SAM files (Table 2). These each accept similar arguments for specifying the coordinate homology and genome region. The “variation scan” command scans each read for the presence or absence of sequence patterns defined by GLUE *Variation* objects [22]. If paired-end read data are supplied, the reads in each pair are processed together. *Variation* objects can encapsulate insertions, deletions, regular expressions and combinations at the nucleotide or amino acid level. In HCV drug resistance this capability may become important. For example, the Magellan-1 trial of the drug pibrentasvir found that the combination of a Methionine at NS5A position 28 with the deletion of the residue at NS5A position 32 was associated with resistance to the drug for HCV subtype 1b [44]. It remains to be seen whether such deletions and combinations occurring as minority variants are clinically relevant but if so, the samReporter offers a means of detecting these.

The commands also allow simple, optional filtering based on Phred base qualities, MAPQ mapping quality and depth. In command outputs, codon numbering is based on the system proposed by Kuiken et al. [18]; nucleotide coordinates both within the SAM file and the mapped location on the master reference are also given. Individual input files may be processed more quickly using parallelisation of command operations across multiple processors. Finally, for paired-end read data, regions where paired reads overlap are not counted twice in command outputs. As part of the GLUE engine, the SAM reporter is implemented in Java, using the Htsjdk library [45] to interpret the SAM format.

The samReporter is delivered as part of the GLUE software package. This study used GLUE version 1.1.33, HCV-GLUE project version 0.1.51 with PHE-HCV-DRUG-RESISTANCE extension version 0.1.21. GLUE is licensed under the open source GNU Affero General Public License version 3.0. and may be installed on Mac OSX, Windows or Linux systems. Documentation specific to the samReporter may be found at: http://tools.glue.cvr.ac.uk/#/deepSequencingData. Documentation for other aspects of GLUE and links to the source code repository can be found on other pages within the same web site.

## 4. Discussion

The results show that, within virus genomes of HCV retreatment patients, linkage between nucleotide variants within a codon location is not a purely theoretical issue. In a small number of cases among UK retreatment patients, such linkage did occur at sites critical for drug resistance. Approaches that do not preserve linkage, such as those encoding variants as ambiguity codes, cannot correctly resolve these cases. How would such a system deal with amino acid residues in the possible set but not in the definite set? If the system is configured not to report such residues, the result is false negative detection of a substitution at NS5A position 30 in three patients, obscuring A30K RAS in two cases. Conversely, if the system reports these residues, the result is false positive detection of the NS5B RAS S282T in three other patients. As far as we are aware, the current study is novel in terms of quantifying the effect of such linkage on resistance detection in real HCV patient data.

The current HCV-GLUE database [27] documents many RASs that combine substitutions at locations within the span of a typical Illumina read. The samReporter can report the presence or absence of these on any read (or read pair for paired-end data) that covers the relevant locations. Future work might consider whether detection of these “combination” RASs at a minority level are of clinical relevance. It would also be of interest to incorporate existing low frequency variant-calling mechanisms from the literature into GLUE. The samReporter and GLUE generally are intended to be useful in both research and clinical contexts. However, even once a SAM file has been generated from a sample, the samReporter only represents one part of the process in terms of drug resistance analysis. The HCV-GLUE system is currently being developed to provide a comprehensive drug resistance report, using samReporter to interpret a SAM file.

Other software, for example the VirVarSeq system [19], calls variants at the codon level but is focused on very low frequency variants. DiversiTools [46] provides frequencies of amino acids on a per read basis but does not link to a standardised coordinate system as is available in the GLUE framework. The geno2pheno[ngs-freq] system [20] directly interprets drug resistance in deep sequencing data. Users must transform their data into a table of nucleotide or codon frequencies and a web-based system then performs interpretation on this table, using a user-defined frequency threshold. In comparison with samReporter, this design facilitates fast transfer over a network since the frequency table is much more compact than a typical SAM or BAM file. However, some information is necessarily lost in the processing, for example the codon frequency table cannot encode linkage beyond a codon location which would be required for example to detect combination RASs.

While the current study applied samReporter to HCV, it can also be used to analyse deep sequencing data for other viruses. In many simple cases, the prerequisites would simply be a nucleotide alignment of alternative target reference sequences and a master reference sequence with coding region annotations. In more complex cases, for example where virus genomes contain ambisense genomes or RNA editing, GLUE and samReporter would need to take account of this. The GLUE samReporter shows that a simple, pragmatic software design can conveniently answer some common questions concerning within-host variation in viral deep sequencing data.

## Figures and Tables

**Figure 1 viruses-11-00323-f001:**
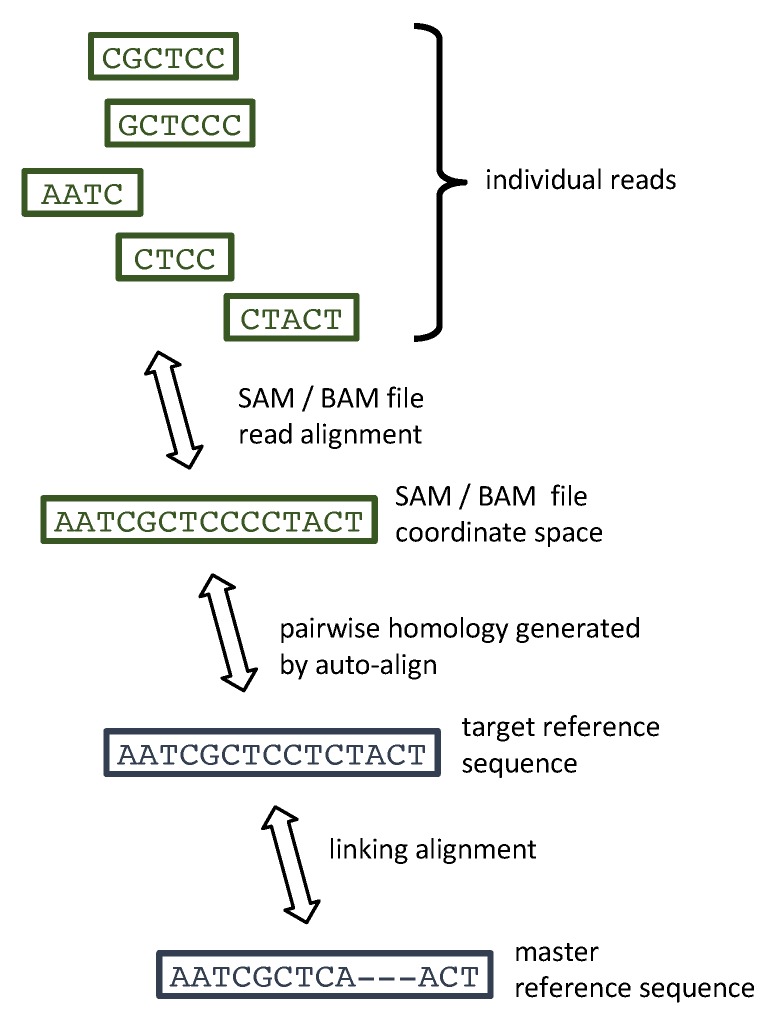
The chain of pairwise homology relationships between reads and the master reference sequence (H77 for HCV), established during the operation of GLUE samReporter.

**Table 1 viruses-11-00323-t001:** Ambiguous resistance-associated locations resolved using GLUE samReporter.

Sequencing	Sample	Subtype	Virus	Codon	Ambiguous	Typical	Possible	Confirmed
Facility	ID		Protein	Location	Triplet	Residue (s)	Residues Set	Residues Set
Glasgow	HCV294	3b	NS5B	282	WSY	S	CST	S
Glasgow	HCV300	3a	NS5A	30	RMG	A	AEKT	AK
PHE	R127	1a	NS5A	24	RSG	K	AGRT	GT
PHE	R164	3a	NS5A	30	RMG	A	AEKT	AK
PHE	R25	4r	NS5B	159	YTM	L	FL	L
PHE	R25	4r	NS5B	282	WSC	S	CST	S
PHE	R36	4r	NS5B	282	WSC	S	CST	S
PHE	R67	1a	NS5A	30	YAW	Q	HQY	QY
PHE	R91	1a	NS5A	28	RYG	M	AMTV	MV
Oxford	7444	3a	NS5A	62	SYA	ST	ALPV	AL

**Table 2 viruses-11-00323-t002:** GLUE samReporter commands.

Command	Description
nucleotide	Generate a table of nucleotide frequencies within a specific genome region.
depth	Generate a table of read depths within a specific genome region.
nucleotide-consensus	Generate a FASTA consensus file, optionally using ambiguity codes.
amino-acid	Generate a table of amino acid residue frequencies within a specific protein-coding region.
codon-triplets	Generate a table of codon frequencies within a specific protein-coding region.
variation scan	Scan for the presence or absence of GLUE *Variations* within reads.
export nucleotide-alignment	Export a specific part of the SAM alignment as a FASTA file.

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
