# Peer review of "Interpreting Viral Deep Sequencing Data with GLUE"

_viruses, 2019, doi:10.3390/v11040323_

Round 1
Reviewer 1 Report
This manuscript used GLUE samReporter, which reported in BMC Bioinformatics, to analysis HCV sam files. This paper explain the application of samReporter for HCV, how about the application for other virus?
Author Response
p.p1 {margin: 0.0px 0.0px 0.0px 0.0px; font: 16.0px Monaco} p.p2 {margin: 0.0px 0.0px 0.0px 0.0px; font: 16.0px Monaco; min-height: 21.0px}
Reviewer 1 indicated: "Extensive editing of English language and style required". We corrected some previously unnoticed
grammar / spelling errors. It was unclear to us what other issues there were. The main author is a native English speaker.
Reviewer 1 comment: "This paper explain the application of samReporter for HCV, how about the application for other virus?"
-- added a paragraph discussing the application of samReporter to other virus species. Lines 354-359
Reviewer 2 comment: "Ln 10 "DAA" treatment in abstract, what is that? please specify it for those unfamiliar readers"
-- abstract updated, Line 10
Reviewer 2 comment: "Ln 31-38. Authors have to differentiate between aligners and assemblers. And as is,
it is kind of a 'mess'. Aligners/mappers and assemblers do different things and are used for different but
complementary purposes. In these sentences, authors confuse the reader because BWA, Bowttie,etc.. are not
assemblers but aligners or read mappers, while SPAdes is a de novo assembler. Please modify these sentences
accordingly to make this clear. Ln 39-40. For instance the SAM file of SPAdes is conceptually different than
the SAM file generated by a read mapper, because SPAdes maps the original reads to the resulting ASSEMBLED
contigs, while with read mappers, reads are aligned against the reference genome, but they do not perform
de novo or a guided assembly. "
-- the relevant paragraph and other parts of the paper have been updated with particular attention
paid to the terminology used. Lines 3, 30-45, 283
Reviewer 2 comment: "Please define what is a “viral cloud”. Authors meant “viral population”?. I think is
better the later form (much more accepted and formal)"
-- the terminology has been updated throughout the manuscript, rephrasing where appropriate. Lines 22, 56, 68, 75.
Reviewer 2 comment: "Methods: Give more details on sequencing methodology, such as kit library prep used,
sequencer model, pair-end, etc ..."
-- more detail has been added in the methods section on sequencing methodologies applied at each site. Lines 209-258
Reviewer 2 comment: "Why not implementing a read trimming step? This is almost mandatory to remove spurious
reads or very low quality reads with Q<30. I don´t really get the point of skipping this because it
introduces more 'noise' that can complicate the downstream analyses of reads such as artifacts in SNVs."
-- trimming and/or quality filtering was in fact applied at an earlier stage for all BAM files,
using different techniques at different sites. Brief details are now given in the Methods section.
Also samReporter commands allow a minimum Phred value to be applied when each command is run.
Read bases which do not meet this threshold do not contribute to the results of the command. A
minimum Phred threshold of 25 was applied both when generating consensus files containing ambiguity
codes and when calculating the frequencies of nucleotide triplets. In both cases this is now mentioned
in the text. Lines 130-131, 168
Reviewer 2 comment: "Remember to add the SRA ID in the last version of the manuscript.
'The deep sequencing data used in this study will be deposited in the NCBI short read archive (SRA) upon publication'."
-- We are working on completing this.
Reviewer 2 comment: "If the ultimate intention of authors with this command program is to aid at the
diagnosis and interpretation of why some HCV variants are resistant at hospitals, my guess is that
facing a command program is not the most friendly presentation. We have to bear in mind that in most
hospitals, seeing around bioinformaticians is not a common thing and thus without a person well
trained in command lines, it would be difficult to implement this tool as a routine."
-- The samReporter and GLUE generally are intended to be useful in both research and clinical contexts. In terms of
resistance analysis of viral sequence data obtained from patients, the samReporter only represents one step in the
process. There are several prior steps required to creating the SAM / BAM file. The samReporter then acts as a tool for
interpreting this in terms we can use, e.g. 50% of reads show a deletion at NS5A position 30. However there are other
parts of the system which would be required, including a database of resistant polymorphisms and an algorithm for
arriving at an overall resistance level for a particular drug. HCV-GLUE includes a prototype drug resistance analysis
system with all these components. HCV-GLUE already has a web-based user interface which can accept consensus
data. A version which accepts deep sequencing data is also available for users to download (e.g. via Docker) and
this can produce drug resistance reports in HTML format from SAM / BAM files using a single command. In future
we may also provide a web-based user interface. The drug resistance system will be the subject of a future article.
We added some text to the Discussion section to explain this. Lines 338-343.
Other changes
Authors: Corrected author's names spelling (Filipe, Mbisa).
Funding: Added description of funding details and disclaimer for Ellie Barnes.
Acknowledgements: Added thanks to the participants and clinicians at NHS sites which contributed samples.
Abstract: "have been suggested" -> "has been suggested".
Spelling mistake in Figure 1 caption.
Figure 1 "master reference" -> "master reference sequence"

Reviewer 2 Report
Authors have designed a bioinformatic tool associate with GLUE program specifically designed to tackle viral sequencing data such as that for HCV. I think the paper is well written and presented. I only have minor concerns.
Ln 10 "DAA" treatment in abstract,what is that? please specify it for those unfamiliar readers
Ln 31-38. Authors have to differentiate between aligners and assemblers. And as is, it is kind of a "mess". Aligners/mappers and assemblers do different things and are used for different but complementary purposes. In these sentences, authors confuse the reader because BWA, Bowttie,etc.. are not assemblers but aligners or read mappers, while SPAdes is a de novo assembler. Please modify these sentences accordingly to make this clear.
Ln 39-40. For instance the SAM file of SPAdes is conceptually different than the SAM file generated by a read mapper, because SPAdes maps the original reads to the resulting ASSEMBLED contigs, while with read mappers, reads are aligned against the reference genome, but they do not perform de novo or a guided assembly.
Please define what is a “viral cloud”. Authors meant “viral population”?. I think is better the later form (much more accepted and formal)
Methods: Give more details on sequencing methodology, such as kit library prep used, sequencer model, pair-end, etc…
Why not implementing a read trimming step? This is almost mandatory to remove spurious reads or very low quality reads with Q<30. I don´t really get the point of skipping this because it introduces more “noise” that can complicate the downstream analyses of reads such as artifacts in SNVs.
Remember to add the SRA ID in the last version of the manuscript. “The deep sequencing data used in this study will be deposited in the NCBI short read archive SRA upon publication”.
Discussion
If the ultimate intention of authors with this command program is to aid at the diagnosis and interpretation of why some HCV variants are resistant at hospitals, my guess is that facing a command program is not the most friendly presentation. We have to bear in mind that in most hospitals, seeing around bioinformaticians is not a common thing and thus without a person well trained in command lines, it would be difficult to implement this tool as a routine.
Author Response

(The authors gave the same response as above.)
